# Factors Controlling the Chromium Isotope Compositions in Podiform Chromitites

**Maria Economou-Eliopoulos [1],\*, Robert Frei [2]**  **and Ioannis Mitsis [1]**

[1] Department of Geology and Geoenvironment, University of Athens, 15784 Athens, Greece; mitsis@geol.uoa.gr

[2] Department of Geosciences and Natural Resource Management, University of Copenhagen, Øster Voldgade 10, 1350 Copenhagen K, Denmark; robertf@ign.ku.dk

\* Correspondence: econom@geol.uoa.gr

**Abstract:** The application of Cr isotope compositions to the investigation of magmatic and post-magmatic effects on chromitites is unexplored. This study presents and compiles the first Cr stable isotope data ($\delta^{53}$Cr values) with major and trace element, contents from the Balkan Peninsula, aiming to provide an overview of the compositional variations of $\delta^{53}$Cr values in ophiolite-hosted chromitites and to delineate geochemical constraints controlling the composition of chromitites. The studied chromitites exhibit $\delta^{53}$Cr values ranging from −0.184‰ to +0.159‰, falling in the range of so-called "igneous Earth" or "Earth's mantle inventory" with values −0.12 ± 0.11‰ to 0.079 ± 0.129‰ (2sd). A characteristic feature is the slightly positively fractionated $\delta^{53}$Cr values of all chromitite samples from Othrys (+0.043 ± 0.03‰), and the occurrence of a wide range of $\delta^{53}$Cr values spanning from positively, slightly negatively to the most negatively fractionated signatures (Pindos, $\delta^{53}$Cr = −0.147 to +0.009‰; Skyros, $\delta^{53}$Cr = −0.078 to +0.159‰). The observed negative trend between $\delta^{53}$Cr values and Cr/(Cr + Al) ratios may reflect a decrease in the $\delta^{53}$Cr values of chromitites with increasing partial melting degree. Alternatively, it may point to processes related to magmatic differentiation, as can be seen in our data from Mikrokleisoura (Vourinos).

**Keywords:** chromium isotopes; chromitites; ophiolites; Balkan Pensula

## 1. Introduction

Chromitites in ophiolite complexes are studied extensively, in particular with respect to mantle-lithospheric slab interactions, post magmatic processes and subduction recycling scenarios [1–8]. Chromitites differ in their major and trace element compositions, including their platinum-group element (PGE) content, and also have been shown to exhibit variations in their chromium isotope signatures. Although the PGE content in chromitites is commonly a few 100 s of ppb, a significant PGE-enrichment, reaching tens of ppm has been reported in high-Cr or high-Al type chromitites from small occurrences, either in Pt and Pd (PPGE; incompatible, with $D$i > 1) or in Os, Ir, Ru (IPGE; compatible, $D$i < 1). Primary magmatic compositional trends recorded in chromite may be obliterated later on during secondary processes, in that Platinum-Group Minenerals (PGM) and silicates may have been substantially modified during subsequent subsolidus re-equilibration and/or by reaction with metasomatic fluids, during extended periods of deformation (including ductile asthenosphere mantle flow), and by shallow crustal, brittle deformation. The factors controlling the trace element distribution in chromitites remain unclear [8–20].

Likewise, limited data on the chromium isotope compositions of chromitites have shown isotopically heavier signatures compared to mantle peridotites [2,21–23]. However, the role of magmatic processes on the Cr isotope system during partial melting and magma differentiation is

still unexplored, and there is still extensive debate on the reasons for the diversity of PGE contents in chromitites [22–24].

Although $\delta^{53}$Cr values of chromitites from the Balkan peninsula are still very limited [2], Cr isotope studies have been performed by several authors in the view to evaluate hexavalent chromium (Cr(VI)) contamination in groundwater and rock leachates [25–30]. For example, Cr isotope signatures indicated that Cr(VI) contamination in groundwater from the Friuli Venezia Giulia Region of Italy is related to the oxidation of trivalent (Cr(III)) deposited as a consequence of past industrial activities [25]. The $\delta^{53}$Cr values and Cr(VI) concentrations in contaminated water from the Evia and Assopos Basins (Greece) are compatible with global data ranges that characterize waters contaminated by both natural processes and anthropogenic activities, and probably delineate potential contamination sources [28]. Also, Cr isotope-based studies have shown oxidative mobilization of Cr(VI) from ultramafic host rocks, and successive back-reduction of the thus mobilized Cr(VI) fractions [29]. Furthermore, the record of a 12-month long time-series of $\delta^{53}$Cr values in run-off from a small serpentinite-dominated catchment in Central Europe, revealed that the Cr(VI) export flux during winter was significantly higher than during the summer [29]. These recent studies and respectively supporting data, in the light of the widespread distribution of ophiolite complexes in Europe and in orogenic zones elsewhere around the globe, emphasize the need for further more detailed studies addressing the variation of $\delta^{53}$Cr values in Cr-bearing rocks and ores.

The ophiolite complexes form an important component of the Tethyan metallogenetic belt, extending from the Mirdita zone in Albania to the Vourinos, Pindos and Othrys complexes of the south zone. These ophiolite complexes host large chromite deposits and smaller chromite occurrences, characterized by high-Cr and high-Al, low- and high-IPGE and PPGE compositions, and by PGE-patterns with both negative and positive slopes [10,14,15,31]. Such a compositional diversity in chromitites may provide an opportunity for in detail studies aimed at a better understanding of the role of magmatic processes on the Cr isotope signatures of the respective chromites [21–24]. In our study we present the first systematic chromium isotope data (expressed as $\delta^{53}$Cr values) of massive chromitites hosted in ophiolite complexes of the Balkan Peninsula, along with trace element contents, including PGE, and combined with scanning electron microscopy/energy-dispersive X-ray spectroscopy (SEM/EDS) identifications. We aim at delineating geochemical constraints that potentially control the elemental composition of these chromitites and their Cr isotope compositions, ultimately with the overall intention to contribute to the origin and genesis of chromite and PGE mineralization and deposits formation.

## 2. Materials and Methods

### 2.1. Preparation and Analysis of the Chromitites

For the purpose of the present study, 2.5 to 3.0 kg amounts of chromitite sample mostly from locations of the Balkan Peninsula (Table 1) were crushed into pieces of approximately $0.3 \times 0.3$ cm with the help of a jaw crusher. The crushed piles were mixed well and divided into four equal portions. Subsequently, two fractions were removed and the remains were remixed again. The procedure was repeated until the final split weighed about 500 g. This split, serving as experimental material, was pulverized to about 100 mesh using an agate mortar. A conventional oxidative alkaline fusion (OAF) was carried out in corundum crucibles; about 1 g of sodium peroxide ($Na_2O_2$) and 0.3 g of $Na_2CO_3$ were added to about 0.1 to 0.3 g of the powdered sample (amount depended on the concentration of Cr in the sample). The sample and flux materials were mixed and molten at temperatures of 700 to 800 °C in a muffle furnace during 10 min. After cooling, the fused cake was extracted from the crucible and transferred into 100 mL volumetric flasks using deionized water [2]. The solutions were finally filtered through 0.45 μm polyamide membrane filters and aliquots of these solutions were then processed for chromium isotope analysis.



### 2.2. Chromium Isotope Analysis

Solutions of samples in the amount which would yield about 1 μg of total chromium were pipetted into 13 mL Savillex Teflon beakers. These aliquots were spiked up with adequate amount of a $^{50}$Cr–$^{54}$Cr double spike so that a sample to spike ratio of ~3:1 (total chromium concentrations) was achieved. The addition of a $^{50}$Cr–$^{54}$Cr double spike of a known isotope composition to a sample before chemical purification allows accurate correction of both the chemical and the instrumental shifts in Cr isotope abundances [1,32]. The mixture was totally evaporated and 1 mL of concentrated aqua regia was subsequently added. After 3 h during which the sample was exposed to aqua regia on a hotplate at 100 °C, the sample was again dried down. The sample was then purified by passing the sample in 0.5 N HCl over an extraction column (BioRad PP columns) charged with 1 mL of 200–400 mesh BioRad AG-50W-X12 cation resin, employing a slightly modified extraction recipe published [33,34]. The Cr yield of this column extraction and purification step is usually ~70%. Samples were loaded onto Re filaments with a mixture of 3 μL silica gel, 0.3 μL 0.5 mol L$^{-1}$ of $H_3BO_3$ and 0.5 μL 0.5 mol L$^{-1}$ of H$_3$PO4. The samples were statically measured on a IsotopX "Phoenix" multicollector thermal ionization mass spectrometer (TIMS) at the Department of Geoscience and Natural Resource Management, University of Copenhagen, at temperatures between 1040 and 1150 °C, aiming for beam intensity at atomic mass unit (AMU) 53.9407 of 1 to 2 Volts. Every load was analyzed five times. Titanium, vanadium and iron interferences with Cr isotopes were corrected by comparing with $^{49}$Ti/$^{50}$Ti, $^{50}$V/$^{51}$V and $^{54}$Fe/$^{56}$Fe ratios. The final isotope composition of a sample was determined as the average of the repeated analyses and reported relative to the certified (standard reference material) SRM 979 standard as

$$\delta^{53}Cr(‰) = [(^{53}Cr/^{52}Cr_{sample}/^{53}Cr/^{52}Cr_{SRM979}) - 1] \times 1000$$

Repeated analysis of 1 μg loads of unprocessed double spiked SRM 979 standard during the duration of the analysis period yielded an average $\delta^{53}$Cr value of 0.05 ± 0.06‰ (n = 12; 2σ; $^{52}$Cr signal intensity at 2 V) on the "Phoenix" TIMS which we consider as a minimum external reproducibility for a sample run in this study, including separation procedure, double spike correction error, and respective internal analytical errors.

### 2.3. Mineral Chemistry and Whole Rock Analyses

Polished sections from chromite deposits of Greece (Table 1) were carbon or gold coated and examined by reflected light microscopy and with a SEM using EDS. SEM images and EDS analyses were carried out at the University of Athens (Department of Geology and Geoenvironment) using a JEOL JSM 5600 instrument, equipped with automated energy-dispersive analysis system ISIS 300 OXFORD, with the following operating conditions: accelerating voltage 20 kV, beam current 0.5 nA, time of measurement (dead time) 50 s and beam diameter 1–2 μm. The following X-ray lines were used: OsM$\alpha$, PtM$\alpha$, IrM$\alpha$, AuM$\alpha$, AgL$\alpha$, AsL$\alpha$, FeK$\alpha$, NiK$\alpha$, CoK$\alpha$, CuK$\alpha$, CrK$\alpha$, AlK$\alpha$, TiK$\alpha$, CaK$\alpha$, SiK$\alpha$, MnK$\alpha$, MgK$\alpha$, ClK$\alpha$. Standards used were pure metals for the elements Cr, Fe, Mn, Ni, Co, Ti and Si, MgO for Mg and Al$_2$O$_3$ for Al. Contents of Fe$_2$O$_3$ and FeO were calculated on the basis of the spinel stoichiometry.

Major and trace elements in massive chromitite samples were determined by inductively coupled plasma mass spectrometry (ICP-MS) analysis, at the ACME Analytical Laboratories Ltd., Vancouver, BC, Canada. The samples were dissolved using a strong multi-acid (HNO$_3$–HClO$_4$–HF) digestion and the residues dissolved in concentrated HCl. Platinum-group element (PGE) analyses were carried out by Ni-sulphide fire-assay pre-concentration technique, using the nickel fire assay technique from large (30 g) samples. This method allows for complete dissolution of samples. Detection limits are 5 ppb for Ru 2 ppb for Os, Ir, Pt, Pd and 1 ppb for Rh and Au. The CDN-PGMS-23 was used as standard.

## 3. A Brief Outline of Characteristics of Ophiolites and Hosted Chromitites

### 3.1. Ophiolites

The geology, petrography, mineral chemistry and geochemistry of ophiolites in the Balkan Peninsula has been a topic of extensive investigation in previous publications [10,35–48]. They are an important component of the Upper Jurassic to Lower Cretaceous Tethyan ophiolite belt, which extends through the Serbian zone of the Dinarides in the north (Mirdita zone) to the Subpelagonian zone (Pindos, Vourinos and Othrys complexes) in the south (Figure 1).

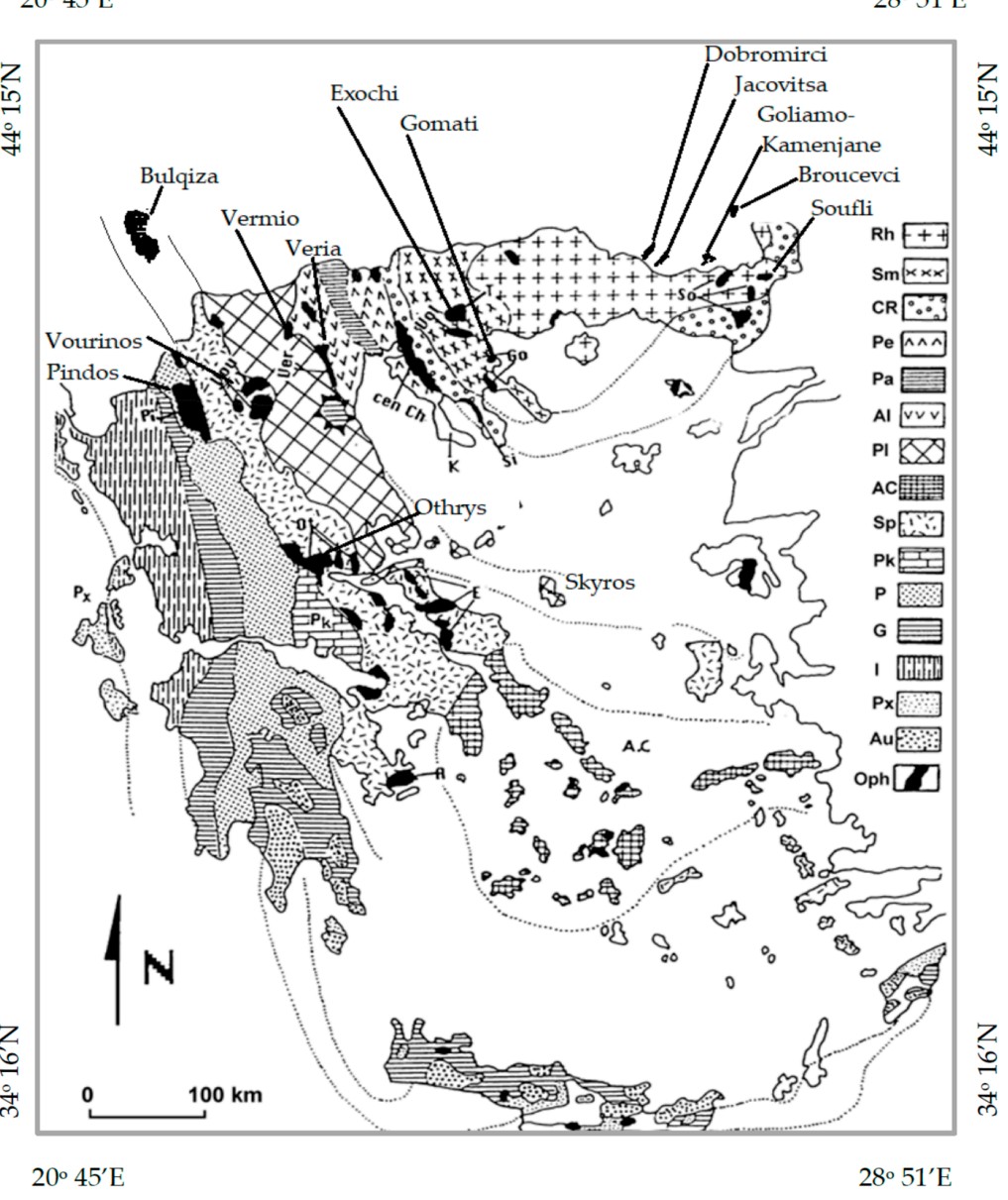

**Figure 1.** Simplified map of the geotectonic zones of Hellenides [35], showing the ophiolite complexes from which chromitite samples were studied. Symbols: Oph = ophiolites; Rh = Rhodope zone; Sm = Serbo–Macedonian Massif; CR = Circun–Rhodope zone; (Pe, Pa, Al) = Axios zone; Pl = Pelagonian zone; AC = Attico–Cucladic zone; Sp = Pindos zone; P = Parnassos–Giona zone; G = Gavrovou–Tripolis zone; I = Ionian zone; Px = Pakson zone.

These ophiolites are characterized by petrological and geochemical features typical of both Mid-Ocean Ridge (MOR) and Supra-Subduction Zone (SSZ)-type associations (fore-arc and back-arc ridges). They are accompanied by minor dunite bodies, and overlain by gabbroic cumulates, mafic dykes, and an extrusive sequence with a compositional basalt range between MOR and island-arc-tholeiitic to boninitic types [35–48]. Recently, ophiolite studies have focused on the importance of intra-oceanic subduction-initiation processes in ophiolite genesis [49]. Relatively small isolated ophiolite masses, of mostly serpentinized dunite and harzburgite, are located at the western margin of the Axios zone (Vermio–Veria), and the Eohellenic Pre-Cretaceous nape, including Skyros Island (Figure 1). Structural and paleomagnetic studies on those ophiolites have revealed widespread heterogeneous deformation and rotation during their original displacement and subsequent tectonic incorporation into continental margins [35–38]. The origin of major Jurassic ophiolite complexes in Greece in a hydrous SSZ environment has been evidenced by the presence of hydrous silicate inclusions, such as amphibole and phlogopite, in chromitites and by the trace and Rare Earth Elements (REE) data on separated orthopyroxenes and clinopyroxenes from harzburgites of these ophiolites [48]. Pyroxenites with variable modal contents of olivine, garnet and spinel, ranging in composition from orthopyroxenite through websterite to clinopyroxenite have been described in the Veria ophiolites [50,51].

## 3.2. Characteristics of Chromitites

Chromitites in the Balkan Peninsula are classified on the basis of the variation in the chromitite tonnage, the composition of chromite, the degree of transformation of ores and the associated ophiolites, and on the association of chromitites with sulphides.

The Othrys complex includes two tectonically separated chromite deposits, namely Eretria (Tsagli) and Domokos, in addition to several other occurrences, including Agios Stefanos, the combined tonnage being approximately 3 Mt of high-Al massive chromite ores occurring in moderately depleted harzburgite [10,35,39,44,45]. A salient feature of the Eretria (Tsagli) chromite deposits is the occurrence of Fe–Ni–Cu-sulphide mineralization with dominant minerals pyrrhotite, chalcopyrite and minor cobalt-bearing pentlandite, hosted in serpentinized harzburgites [52]. Massive sulphide mineralization occurs at the peripheral parts of podiform chromite bodies in association with chromitite and magnetite (Figure 2). Chromite is often characterized by brittle deformation and by sulphide-silicate veins (Figure 2d,e). Texture and geochemical characteristics, including PGE content, flat chondrite-normalized PGE-patterns, and very low partition coefficients for Ni and Fe between olivine and sulphides are inconsistent with sulphides having been in equilibrium with Ni-rich host rocks, at magmatic temperature [52,53]. In addition, it has been documented [54] that in the Othrys and Vourinos ophiolite complexes massive and disseminated chromitites host considerable amounts of methane in micro-fractures and in porous serpentine- or chlorite-filled veins.

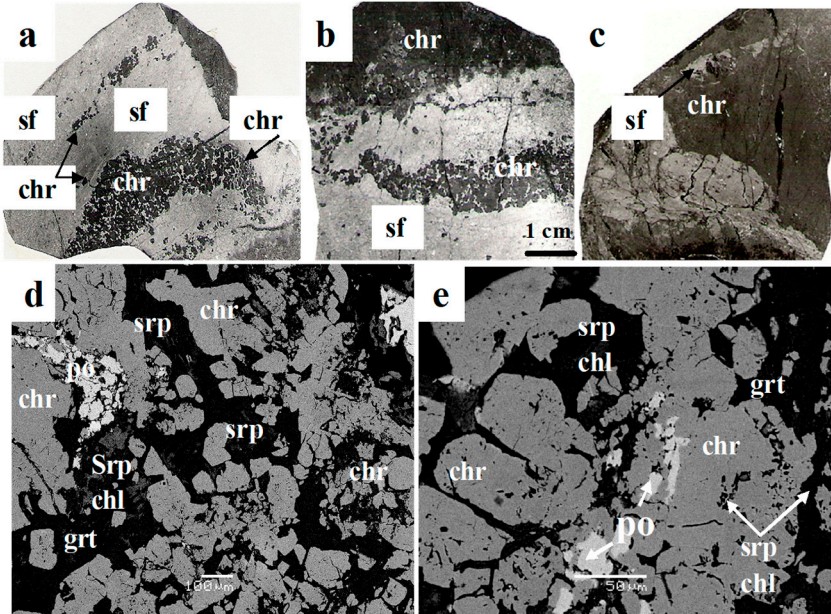

**Figure 2.** Photographs showing the texture of chromitite associated with sulphides and magnetite from the peripheral parts of podiform chromitite bodies at Eretria. Chromite occurs interstitially in sulphide rich portions (**a**,**b**) and sulphides occur as veins and irregular masses within zones dominated by chromite (**b**,**c**). Back-scattered electron images from massive chromitite samples from central parts of podiform bodies, showing fragmented chromite and evidence for the existence of sulphides (**d**,**e**) even distant from the peripheral parts of the chromitite bodies. Abbreviations: chr = chromite; sf = sulphides; po = pyhhrotite; srp = serpentine; chl = chlorite; grt = garnet. Scale bar: 1 cm is the same for all hand samples.

Chromite ores in the Vourinos complex occur in tectonite and cumulate sequences. Only those in the tectonites are under exploitation. All chromite ores, with a tonnage estimated to be approximately 10 Mt of high-Cr ore are found within dunite bodies or dunite envelopes in harzburgite, but there is no systematic relation between size of the dunite body and that of the ore body. The sizes of the ore bodies vary widely and contain all textural types (massive, schlieren, banded, disseminated and nodular), but usually a single type dominates. High-temperature deformation, superimposed on primary magmatic textures is very common [42,47]. The mantle sequence of the Pindos ophiolite complex resembles that of Vourinos in the presence of extensive and highly depleted harzburgite but, in contrast to Vourinos, there is only a limited number of small chromite occurrences (low potential for exploitation).

The chromitite occurrences in the Pindos ophiolite complex are small (a few tens of meters × a few tens of centimeters) and cover all textural types (massive, schlieren, banded, disseminated and nodular). Chromitites occur within completely serpentinized and weathered, intensively deformed dunite–harzburgite blocks, as a result of strong plastic and brittle deformation that was superimposed on primary magmatic textures. The chromitites are mostly fine- to medium-grained, and consist of aggregates of fractured chromite accompanied by chlorite and traces of tremolite. Primary olivine is preserved only in the form of inclusions within chromian spinel, while abundant remnants of base metal sulphides (BMS), now preserved as alloys, occur both as inclusions and interstitial phases in the silicate matrix [10,18,19]. Chromitites throughout the Pindos complex are of high-Cr and high-Al compositions, often in a spatial association with each other [45,55–57]. The most salient feature of the Pindos chromitites is the PPGE-enrichment in those bodies occurring in the area of Korydallos, with PGE$_{total}$ concentrations reaching 7 ppm [43–45] and reportedly up to 29 ppm [18].

Relatively small massive chromitite bodies are located at the Achladones area on the island of Skyros. They are of high-Al type (average Cr/(Cr + Al) ratio of unaltered chromite is 0.56 while the Mg/(Mg + Fe$^{2+}$) ratio is 0.64) and contain elevated PGE contents, up to 3 ppm PGE$_{total}$. However,

both high-Cr and high-Al types have been reported from Skyros [46,47,58]. Furthermore, SEM/EDS investigations of Au-coated polished sections of chromitite from Skyros revealed the presence of graphite (Figure 3).

Ophiolites associated with the Serbomacedonian massif (Gomati and Exochi) and Rhodope massif including the ophiolites Soufli–Tsoutoura (Greece), Dobromirci, Jacovitsa, Broucevci and Goliamo–Kamenjane (Bulgaria) (Figure 1) are completely serpentinized, locally sheared and metamorphosed to antigorite–tremolite and/or talc schists, and they are all small (a few thousand tons of ore). A characteristic feature of those chromitite occurrences is the common spatial association of high-Cr and high-Al ores, and the negative slope of the PGE patterns, comparable to those for podiform chromitites in ophiolites elsewhere [59–61]. The area of Ceruja is part of the western Bulqiza ophiolite complex as part of the Mirdita zone (Albanides), and includes harzburgitic to lherzolitic types [62]. Although more than 40 Mt of chromitites have been located in the uppermost part of the mantle harzburgites throughout the Bulqiza complex, in the area of Ceruja there are only small (a few thousand) chromite occurrences.

There is a wide compositional variation in chromite throughout the Bulqiza complex, with Cr/(Cr + Al) ranging from 0.46 to 0.86 and Mg/(Mg + Fe$^{2+}$) ratios ranging from 0.46 to 0.69. The Cr/Cr + Al) ratio in studied samples from the area of Ceruja is restricted to 0.43–0.56, their PGE content is low and the PGE-patterns show a negative slope, similar to other chromitites hosted in ophiolite complexes [62]. However, in the western Bulqiza massif, mineralization occurring as disseminations in dunite is accompanied by chromite in the transition zone and lower cumulates sequence of the complex and elevated PGE contents, up to 9000 ppb ΣPGE [62–64].

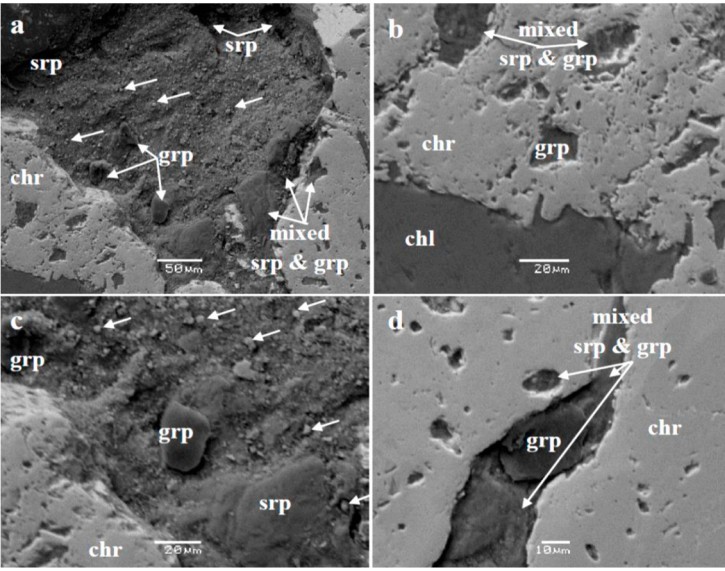

**Figure 3.** Back-scattered electron images (Au-coated polished sections) of strongly fragmented chromitite from Skyros Island, showing microstructural inclusions of graphite. Partial replacement of serpentine flakes by graphite (**a**,**b**,**d**); Overgrowth of fine graphite nodules (white arrows) are revealed on unpolished parts of polished sections (**a**,**c**); (**f**) Dissolution pits in silicate inclusions of chromite are filled with graphite (**b**,**d**). Abbreviations: grp = graphite-like chr = chromite; srp = serpentine; chl = chlorite.

## 4. Results

### 4.1. Distribution of Selected Trace Elements

Selected trace element contens of Ni, Co, Mn, Zn, V, Ga Ti, and platinum-group elements, which are from previous studies, are contained in Table 1.

**Table 1.** Chromium isotope ($\delta^{53}$Cr) and selected trace element data for chromitites from the Balkan Peninsula.

| No | Location | $\delta^{53}$Cr | SEM/EDS | | PGE Content (ppb) | | | | | | | | | Trace Element Content (ppm) | | | | | | | wt % | wt % | wt % |
|---|---|---|---|---|---|---|---|---|---|---|---|---|---|---|---|---|---|---|---|---|---|---|---|
| | | (‰) | Cr≠ | Mg≠ | Os | Ir | Ru | Rh | Pt | Pd | ΣPGE | Pd/Ir | Pt/Pt * | Ni | Co | V | Zn | Mn | Ga | Sc | Fe | Ca | Ti |
| | Othrys (D), GR | | | | | | | | | | | | | | | | | | | | | | |
| 1 | Eretria | 0.023 | 0.57 | 0.63 | 22 | 13 | 67 | 2 | 10 | 1 | 115 | 0.08 | 2.26 | 1600 | 220 | 900 | 340 | 1250 | 34 | 6 | 9.6 | <0.1 | 0.03 |
| 2 | Eretria | 0.036 | 0.56 | 0.66 | 25 | 14 | 25 | 3 | 4 | 1 | 72 | 0.07 | 0.74 | 1400 | 210 | 1100 | 430 | 990 | 32 | 6 | 10 | 0.2 | 0.03 |
| 3 | Domokos | 0.025 | 0.53 | 0.74 | 15 | 11 | 30 | 5 | 24 | 2 | 87 | 0.18 | 2.43 | 1600 | 200 | 960 | 310 | 1300 | 32 | 7 | 9.8 | <0.1 | 0.04 |
| 4 | Domokos | 0.088 | 0.5 | 0.73 | 20 | 10 | 35 | 5 | 8 | 11 | 89 | 1.1 | 0.34 | 1200 | 220 | 900 | 400 | 2500 | 33 | 6 | 9.6 | <0.1 | 0.04 |
| | Vourinos (D), GR | | | | | | | | | | | | | | | | | | | | | | |
| 5 | Kondro | | 0.77 | 0.65 | 26 | 36 | 55 | 17 | 4 | 2 | 140 | 0.06 | 0.22 | 2000 | 240 | 500 | 260 | 1560 | 14 | 6 | 8.1 | <0.1 | 0.04 |
| 6 | Voidolakos | −0.12 | 0.81 | 0.62 | 14 | 17 | 80 | 12 | 6 | 6 | 135 | 0.35 | 0.23 | 1580 | 210 | 560 | 550 | 1540 | 14 | 6 | 7.9 | <0.1 | 0.04 |
| 7 | Kissavos | −0.093 | 0.81 | 0.66 | 14 | 11 | 49 | 7 | 4 | 7 | 92 | 0.55 | 0.55 | 1900 | 200 | 400 | 300 | 1000 | 13 | 6 | 8.7 | <0.1 | 0.04 |
| 8 | Kissavos | −0.078 | 0.56 | 0.72 | 4 | 3 | 11 | 3 | 3 | 6 | 30 | 0.54 | 0.23 | 1800 | 170 | 780 | 360 | 850 | 27 | 6 | 9.2 | <0.1 | 0.03 |
| 9 | Mikrokleisoura | −0.056 | 0.78 | 0.61 | 30 | 11 | 40 | 16 | 8 | 4 | 109 | 0.63 | 0.26 | 1800 | 180 | 620 | 280 | 950 | 14 | 8 | 7.8 | <0.1 | 0.05 |
| 10 | Mikrokleisoura | −0.032 | 0.61 | 0.52 | 50 | 8 | 42 | 18 | 8 | 6 | 132 | 0.75 | 0.96 | 1000 | 230 | 1000 | 490 | 1760 | 20 | 12 | 15 | 0.5 | 0.1 |
| | Pindos, Low PGE | | | | | | | | | | | | | | | | | | | | | | |
| 11 | P Dako's mine | −0.03 | 0.52 | 0.63 | 4 | 6 | 30 | 1 | 4 | 6 | 51 | 1.0 | 0.52 | 1300 | 140 | 600 | 400 | 760 | 23 | 5 | 7.8 | <0.1 | 0.04 |
| 12 | Korydallos | 0.009 | 0.48 | 0.6 | 40 | 20 | 33 | 5 | 25 | 20 | 143 | 1.0 | 0.8 | 1500 | 260 | 710 | 410 | 1200 | 32 | 5 | 9.8 | <0.1 | 0.08 |
| 13 | Kampos Despoti | −0.096 | 0.81 | 0.58 | 30 | 18 | 50 | 3 | 76 | 4 | 181 | 0.22 | 7.01 | 1450 | 290 | 560 | 490 | 1470 | 17 | 5 | 10 | <0.1 | 0.03 |
| 14 | Korydallos | −0.149 | 0.42 | 0.71 | 8 | 3 | 11 | 1 | 75 | 19 | 117 | 6.33 | 5.5 | 1500 | 260 | 710 | 520 | 1300 | 34 | 5.1 | 7.8 | <0.1 | 0.09 |
| | Pindos, High-PPGE | | | | | | | | | | | | | | | | | | | | | | |
| 15 | Korydallos | −0.098 | 0.68 | 0.57 | 14 | 11 | 57 | 13 | 3460 | 1660 | 5215 | 151 | 7.53 | 1550 | 230 | 510 | 450 | 1200 | 30 | 5.7 | 7.8 | <0.1 | 0.07 |
| 16 | Korydallos | 0.062 | 0.67 | 0.57 | 47 | 49 | 55 | 104 | 3020 | 600 | 3875 | 12.2 | 3.86 | 750 | 270 | 760 | 520 | 1280 | 16 | 6.8 | 9.7 | <0.1 | 0.07 |
| 17 | Korydallos | −0.102 | 0.69 | 0.49 | 62 | 47 | 80 | 112 | 1460 | 337 | 2098 | 7.2 | 2.4 | 720 | 240 | 760 | 620 | 1340 | 23 | 6.7 | 10 | <0.1 | 0.08 |
| 18 | Korydallos | −0.06 | 0.4 | 0.72 | 88 | 74 | 130 | 142 | 1720 | 525 | 2679 | 7.1 | 2.01 | 630 | 240 | 790 | 510 | 1300 | 25 | 6.8 | 9.9 | <0.1 | 0.09 |
| 19 | Korydallos | −0.086 | 0.4 | 0.72 | 45 | 38 | 82 | 78 | 4580 | 1300 | 6123 | 34.2 | 4.6 | 1630 | 240 | 580 | 460 | 1400 | 32 | 5.4 | 7.3 | <0.1 | 0.07 |
| | Pindos, High-IPGE | | | | | | | | | | | | | | | | | | | | | | |
| 20 | Milia | −0.147 | 0.82 | 0.42 | 150 | 320 | 350 | 82 | 150 | 7 | 1059 | 0.02 | 2.00 | 1600 | 340 | 980 | 1160 | 2000 | 15 | 5 | 12 | 0.1 | 0.01 |
| | Skyros, high IPGE | | | | | | | | | | | | | | | | | | | | | | |
| 21 | Achladones | −0.078 | 0.61 | 0.64 | 140 | 480 | 1200 | 160 | 280 | 39 | 2300 | 0.08 | 1.13 | 1300 | 250 | 1200 | 540 | 1300 | 34 | 5 | 11 | <0.1 | 0.04 |
| 22 | Achladones | 0.117 | 0.55 | 0.69 | 60 | 40 | 300 | 13 | 23 | 28 | 464 | 0.7 | 0.39 | 1600 | 200 | 1000 | 400 | 1650 | 36 | 5 | 9.4 | <0.1 | 0.04 |
| 23 | Achladones | 0.159 | 0.58 | 0.51 | 13 | 31 | 159 | 7 | 20 | 21 | 251 | 0.67 | 0.53 | 1500 | 240 | 870 | 450 | 1600 | 40 | 5 | 9.6 | <0.1 | 0.04 |
| | Skyros, Low PGE | | | | | | | | | | | | | | | | | | | | | | |
| 24 | Ag. Ioannis | −0.111 | 0.75 | 0.69 | 30 | 15 | 85 | 6 | 4 | 5 | 145 | 0.33 | 0.23 | 1250 | 220 | 640 | 420 | 2150 | 14 | 6 | 11 | <0.1 | 0.08 |
| 25 | Parapisti | −0.084 | 0.79 | 0.69 | 25 | 20 | 90 | 5 | 3 | 2 | 145 | 0.1 | 0.3 | 1200 | 200 | 620 | 400 | 1900 | 11 | 5 | 10 | <0.1 | 0.05 |
| 26 | Ceruja | −0.051 | 0.43 | 0.69 | 3 | 7 | 18 | 3 | 25 | 7 | 63 | 1 | 1.74 | 2640 | 170 | 610 | 300 | 960 | | | | | |
| 27 | Ceruja | −0.129 | 0.52 | 0.58 | 3 | 7 | 15 | 2 | 28 | 9 | 64 | 1.3 | 2.1 | 1640 | 160 | 1030 | 420 | 1210 | | | | | |
| 28 | Ceruja | −0.047 | 0.56 | 0.65 | 3 | 7 | 18 | 3 | 25 | 7 | 63 | 1 | 1.74 | 2150 | 160 | 990 | 340 | 1100 | | | | | |

**Table 1.** *Cont.*

| No | Location | δ⁵³Cr | SEM/EDS | | PGE Content (ppb) | | | | | | | | | Trace Element Content (ppm) | | | | | | | wt % | wt % | wt % |
|---|---|---|---|---|---|---|---|---|---|---|---|---|---|---|---|---|---|---|---|---|---|---|---|
| | | (‰) | Cr≠ | Mg≠ | Os | Ir | Ru | Rh | Pt | Pd | ΣPGE | Pd/Ir | Pt/Pt * | Ni | Co | V | Zn | Mn | Ga | Sc | Fe | Ca | Ti |
| | Vermio-Veria | | | | | | | | | | | | | | | | | | | | | | |
| 29 | Veria | −0.075 | 0.75 | 0.62 | 70 | 15 | 100 | 9 | 3 | 2 | 199 | 0.13 | 0.23 | 1500 | 200 | 650 | 700 | 2800 | | | | | |
| 30 | Vermio | −0.076 | 0.7 | 0.65 | 30 | 20 | 55 | 4 | 4 | 31 | 144 | 0.05 | 0.12 | 880 | 170 | 700 | 500 | 1400 | | | | | |
| 31 | Rhodope massif, GR | −0.145 | 0.75 | 0.66 | 70 | 15 | 100 | 9 | 3 | 2 | 199 | 0.13 | 0.23 | 1770 | 150 | 550 | 300 | 1800 | | | | | |
| | Rhodope massif, Bu | | | | | | | | | | | | | | | | | | | | | | |
| 32 | Soufli | −0.037 | 0.7 | 0.62 | 20 | 25 | 85 | 7 | 7 | 5 | 150 | 0.2 | 0.38 | 1260 | 210 | 610 | 360 | 1850 | | | | | |
| 33 | Tsoutoura | −0.033 | 0.63 | 0.65 | 25 | 6 | 40 | 2 | 3 | 6 | 82 | 1.00 | 0.28 | 1430 | 220 | 670 | 360 | 1270 | | | | | |
| 34 | Dobromirci | −0.184 | 0.65 | 0.68 | 16 | 32 | 120 | 6 | 4 | 14 | 190 | 0.44 | 0.14 | 520 | 160 | 590 | 230 | 2300 | | | | | |
| 35 | Goliamo Kamenjane | −0.099 | 0.6 | 0.62 | 3 | 6 | 14 | 3 | 3 | 1 | 30 | | | 2000 | 780 | 500 | 3700 | 7500 | | | | | |
| 36 | Broucevci | −0.147 | 0.47 | 0.72 | 5 | 13 | 23 | 4 | 2 | 12 | 60 | 0.92 | 0.09 | 800 | 180 | 600 | 370 | 2500 | | | | | |
| 37 | Jacovitsa | −0.097 | 0.75 | 0.61 | 22 | 42 | 53 | 7 | 3 | 2 | 129 | 0.05 | 0.26 | 790 | 240 | 650 | 400 | 3500 | | | | | |
| | SMM, GR | | | | | | | | | | | | | | | | | | | | | | |
| 38 | Exochi | 0.02 | 0.76 | 0.62 | 23 | 24 | 73 | 6 | 5 | 7 | 140 | 0.29 | 0.25 | 1240 | 240 | 670 | 400 | 1850 | | | | | |
| 39 | Exochi | −0.057 | 0.75 | 0.6 | 25 | 10 | 70 | 4 | 4 | 6 | 119 | 0.6 | 0.26 | 1030 | 150 | 1300 | 300 | 1270 | | | | | |
| 40 | Gomati | 0.084 | 0.49 | 0.67 | 9 | 20 | 60 | 6 | 5 | 4 | 104 | 0.25 | 0.33 | 1050 | 330 | 660 | 350 | 2700 | | | | | |

* Present data; PGE data from previous publications [24,34,43,44,50]. Symbols: Cr# = Cr/(Cr + Al); Mg# = (Mg + Fe$^{2+}$).

The rare earth elements La, Ce, Nd, Sm, Yb, Pr, Eu, Gd, Tb, Dy, Ho, Er, Tm and Lu as well as U, Th, Cs, Sb, Y, Zr, Ge, W, Mo, Sc, Pb, Li, Sr and K were lower than the detection limits in detail specified in the section of the analytical methods, except for La and Ce contents in the chromitite samples from Othrys and Skyros island, which are slightly higher than the respective detection limit. The best defined inverse correlations are those between the Cr/(Cr + Al) ratio and V (r = −0.80 to −0.85) for chromitites from Vourinos, Pindos (low PGE) and Skyros, and between the Cr/(Cr + Al) ratio (Figure 4a), and Ga (r = −0.90 to −0.96) for chromitites from Vourinos, Pindos (low PGE) and Skyros and for high-PGE chromitites (r = −0.65; Figure 4b).

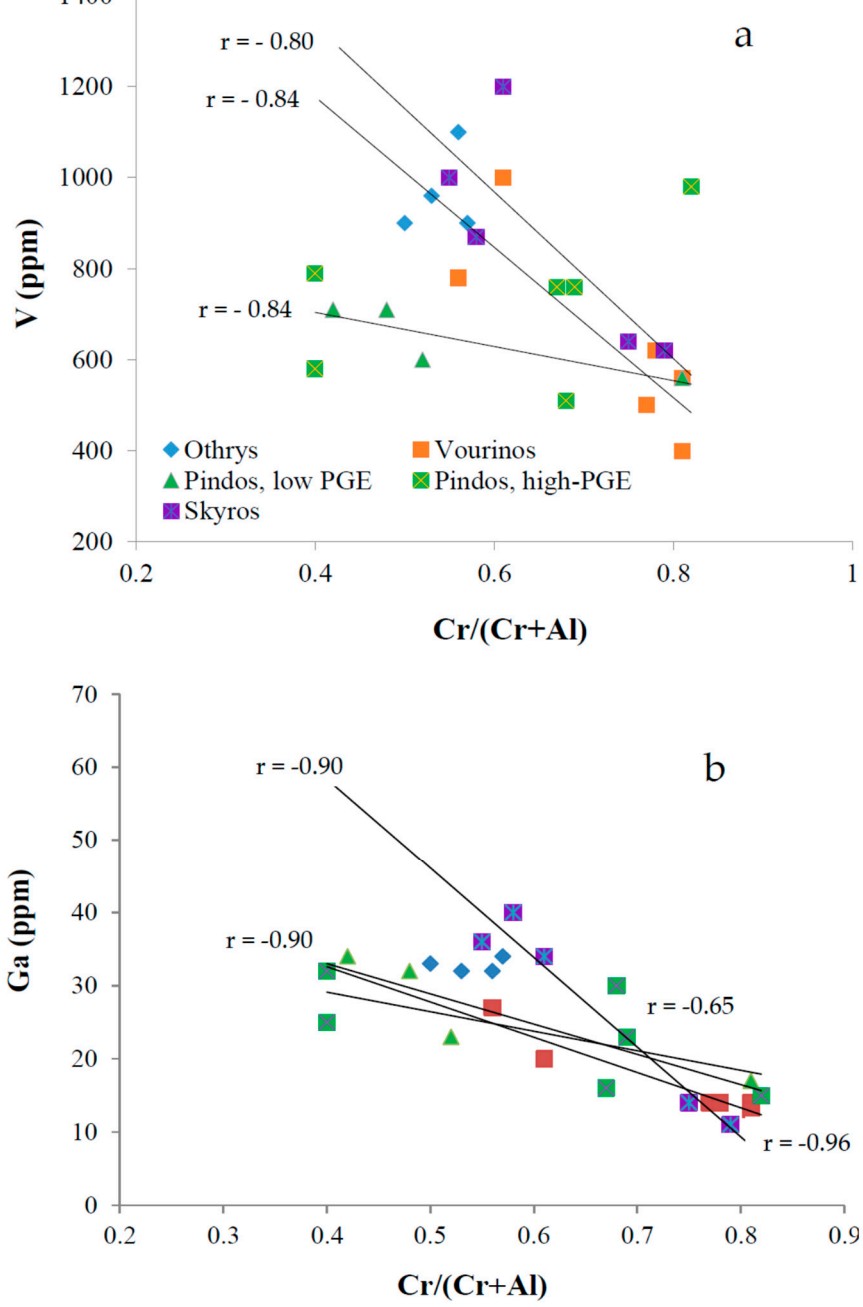

**Figure 4.** Plots of the Cr/(Cr + Al) ratio versus V and Ga for chromites from the Balkan Peninsula. The best pronounced negative correlations are those between Cr/(Cr + Al) ratio and V for chromitites from Vourinos, Pindos (low PGE) and Skyros (**a**), and Ga for chromitites from Vourinos, Pindos (low PGE) and Skyros (**b**). Data from Table 1.

### 4.2. Chromium Isotope Results

The analysed chromitites, representative of a wide range in their major element (high-Cr and high-Al) and PGE composition, show a relatively wide range of chromium isotope compositions, expressed as $\delta^{53}$Cr values, even in spatially related occurrences, throughout the entire metallogenic belt. The $\delta^{53}$Cr values range from −0.184‰ (in the Bulgarian Rhodope massif) to +0.159‰ on Skyros Island (Figure 5; Table 1). The lower values (−0.246‰) are comparable to those having been reported by [2]. $\delta^{53}$Cr as low as −0.21‰ are compatible with the $\delta^{53}$Cr isotope range for mantle-derived rocks [1], while the more positively fractionated values of up to +0.159‰ exceed those values around +0.068‰ reported for global chromitites from layered intrusions and podiform-type occurrences [2].

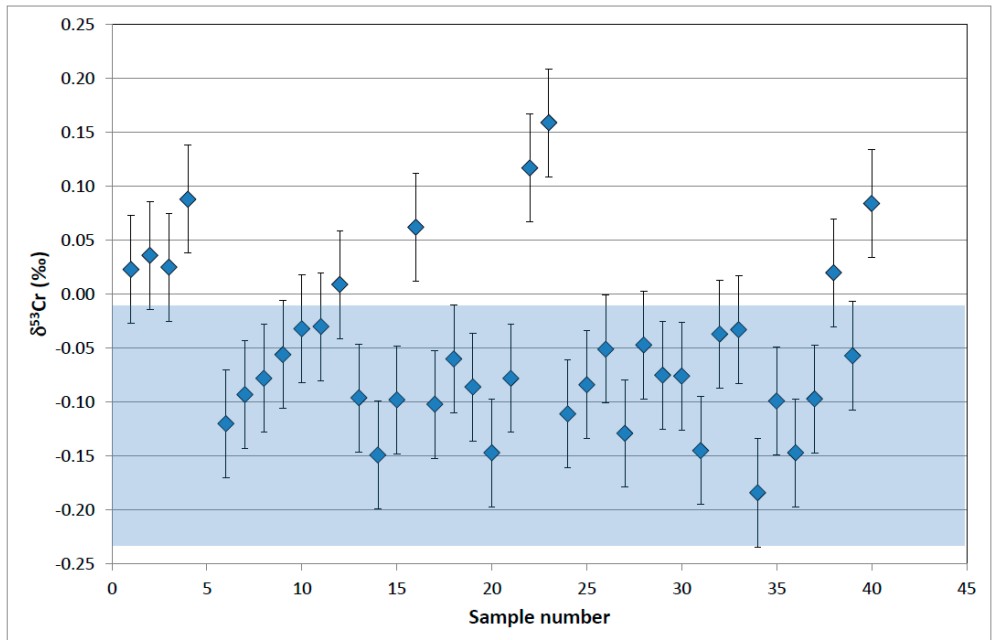

**Figure 5.** $\delta^{53}$Cr values of the chromitites (n = 40) from the Balkan Peninsula (Table 1) studied herein. The blue coloured band displays the range of high-temperature magmatic signatures as has been defined by [1]. Errors are 2sd. The reproducibility of the (standard reference material) SRM 979 standard under similar measuring conditions is at +/−0.06‰. The horizontal axis corresponds to the number of analyzed samples 1–40 (Table 1).

Along with the $\delta^{53}$Cr values of the chromitites, selected major and trace element compositions reported in previous publications are given (Table 1) in order to delineate potential geochemical constraints that control their composition and the Cr isotope compositions. It is remarkable that all chromite samples from Othrys and certain ores from Skyros exhibit positive $\delta^{53}$Cr values, while chromitites from the Serbo-macedonian-Rhodope massifs and Pindos show a wide range from positive, slightly negative to the most negative $\delta^{53}$Cr values (Figures 5 and 6; Table 1). Also, the results do not show any correlation trend either between total PGE content or Mg/(Mg + Fe$^{2+}$) ratio and $\delta^{53}$Cr values. There is a slightly negative trend between $\delta^{53}$Cr values and Cr/(Cr + Al) ratio, which is better pronounced for the chromitite samples from Othrys and Vourinos (Figure 6a) and a positive trend between $\delta^{53}$Cr values and Ga content (Figure 6b).

The Pd/Ir values for chromitites from the Balkan Peninsula are plotted *versus* normalized values (Pt/Pt* = (Pt/8.3)/(Rh/1.6)(Pd/4.4) (Figure 7, [65])). The negative Pt/Pt* values (Pt/Pt* < 1) and low Pd/Ir ratios are a characteristic feature of IPGE-elevated chromitites from Skyros and of low-PGE chromitites, in contrast to PPGE-elevated ones from the Pindos complex (Figure 7; Table 1).

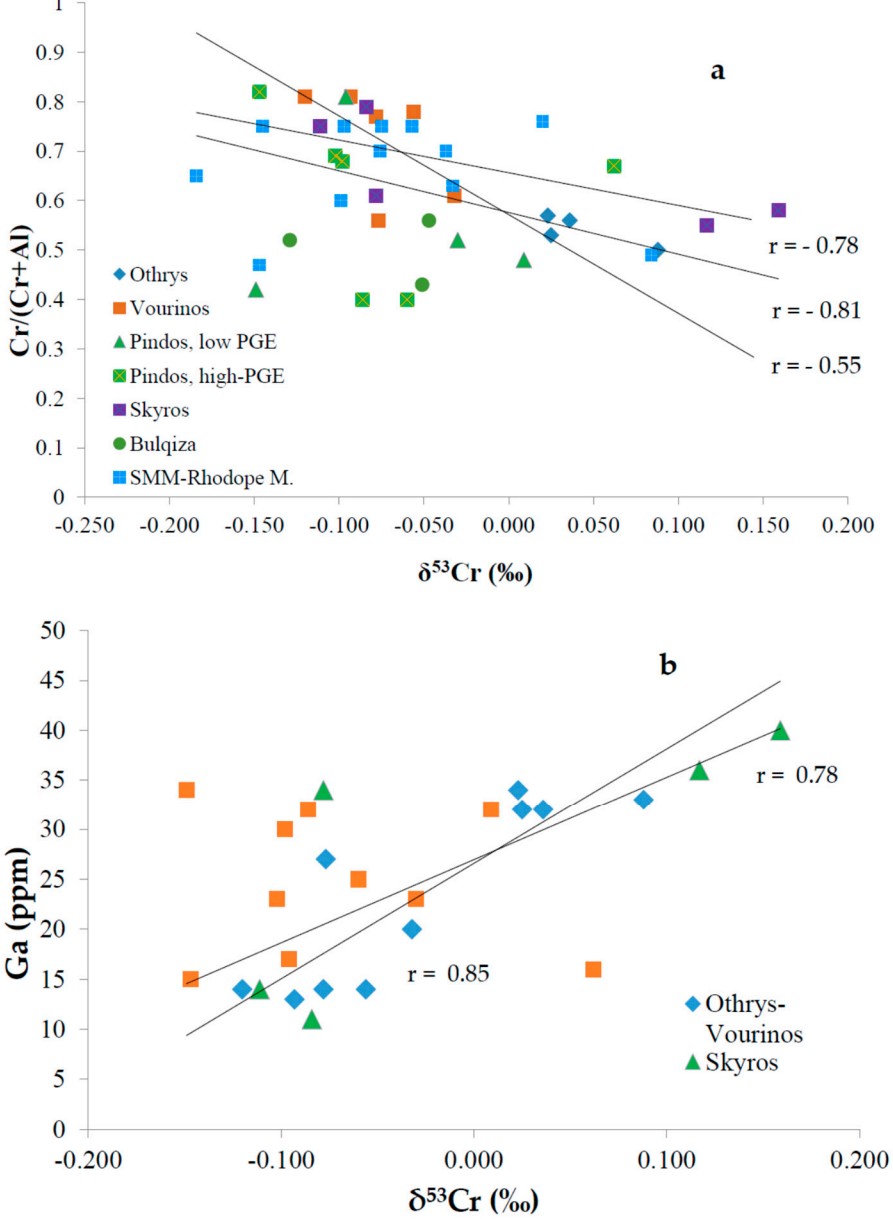

**Figure 6.** Plots of $\delta^{53}$Cr values *versus* Cr/(Cr + Al) ratios and Ga for certain chromitites from the Balkan Peninsula. There is a good negative trend between $\delta^{53}$Cr values and Cr/(Cr + Al) ratios for chromitites from Othrys and Skyros (**a**), and a positive trend between $\delta^{53}$Cr values and Ga contents for chromitites from Skyros and Vourinos–Othrys (**b**). Data from Table 1.

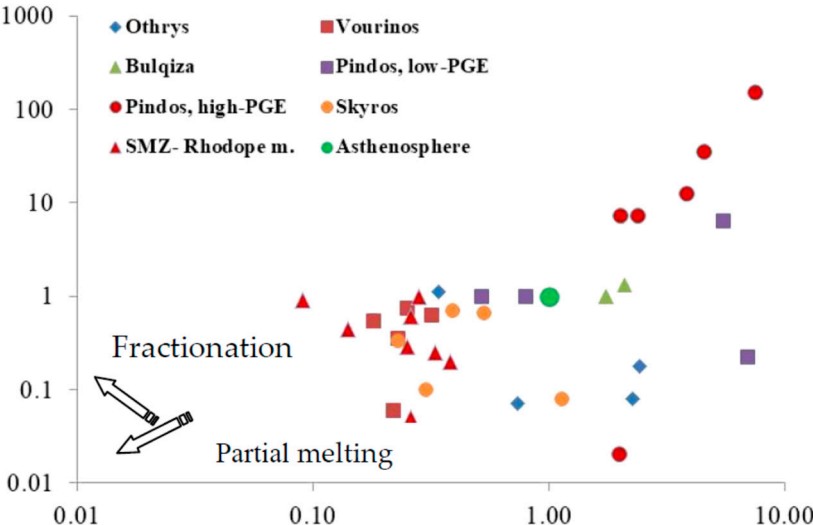

**Figure 7.** Plots of Pd/Ir *versus* Pt/Pt* normalized values [(Pt/Pt* = (Pt/8.3)/(Rh/1.6) × (Pd/4.4)] after [65] of chromitites studied herein, grouped by locality. Green circle = Asthenosphere mantle, calculated by [66]. Data from Table 1.

## 5. Discussion

Geochemical compositions and mineral chemistry characteristics of peridotites and chromitites from Greece have shown that the partial melting degree of primitive mantle and the hydrous nature of parent magma are major factors that control the compositional variations in them [10,18,19,42,45,56,57]. In addition, the known widespread geochemical heterogeneities in the bulk-rock compositions of chromitites from the Balkan Peninsula (Table 1) would be inconsistent with their evolution through simple partial melting processes, and suggest that their original geochemical compositions may have been modified by subsequent, post-magmatic processes taking place at shallow mantle depths [8,66–69].

### 5.1. Potential Pathways Contributing to Chromium Isotope Fractionation

Chromium occurs in different oxidation states in nature, with relatively immobile Cr(III) and very soluble oxidized Cr(VI) being the most abundant species [70]. Redox processes are accompanied by significant isotope fractionation, also in systems where Cr(VI) is transformed to Cr(III) [71]. During reduction, the lighter isotopes are preferentially reduced, resulting in an enrichment of $^{53}$Cr relative to $^{52}$Cr values in the remaining Cr(VI) pool. This enrichment is measured as the change in the ratio of $^{53}$Cr/$^{52}$Cr, and is expressed as $\delta^{53}$Cr values in units permil (‰) relative to a standard [32]. In the following we consider potential pathways for changes in redox conditions that might affect the Cr isotope signatures and their changes in magmatic to post-magmatic processes: (1) measured $\delta^{53}$Cr signatures in mantle chromitites reflect primary magmatic compositional variations, (2) oxidative fluids may affect solubility of Cr-bearing minerals and release of Cr(VI) during the formation of ferrian-chromite, (3) reductive fluids may cause reduction of Cr(VI) back to Cr(III), deposition of graphite and Fe–Cr alloys and (4) weathering of Cr-bearing minerals [19–21,25,60–64], and related dissolution/re-precipitation processes potentially having a control on the bulk $\delta^{53}$Cr signatures of chromitites. Based on the observation that bulk silicate earth (BSE) carries a slightly lighter $\delta^{53}$Cr signature compared to that of mantle-derived chromitites, it has been suggested that small but discernible high-temperature Cr isotope fractionations might occur within the Earth's mantle, potentially imparted during fractional crystallization and/or partial melting of mantle sources [1].

#### 5.1.1. Magmatic Processes

It has been established that high-Cr and high-Al podiform chromitites are derived from melts formed by high and low degrees of partial melting in the upper mantle, respectively [3,13,14,72].

Considering the data distribution in the scatter-diagram of Pd/Ir versus the Pt/Pt* ratios for chromitites from the Balkan Peninsula, it appears that chromitites from the large deposits (Vourinos and Othrys) exhibit Pd/Ir < 1 and Pt/Pt* < 1, suggesting partial melting of already depleted mantle [65], and an increasing partial melting trend from high-Al toward high-Cr deposits (Table 1; Figure 7). In addition, the relatively high Pd/Ir ratio, that is considered to indicate the fractionation degree of the PGE [66], provides evidence for fractionation to some extent of parent magmas for the PPGE-elevated chromitites from Pindos compared to the IPGE-elevated chromitites from Skyros and the Milia area of Pindos (Table 1; Figure 7).

The closer inspection of the Cr isotope data from chromitites studied herein, ranging from negative (−0.184‰) to positive (+0.159‰) $\delta^{53}$Cr values (Table 1; Figures 5 and 6), may provide an opportunity for a better understanding of the role of magmatic or/and post-magmatic processes on the Cr isotope signatures of chromitites [2,21–23]. A comparison of the Cr isotope data for chromitites from our study (Table 1) with published values from other chromitites reveals a significant overlap in the $\delta^{53}$Cr data. Our lowest $\delta^{53}$Cr signature (−0.184‰) is slightly higher than the most negatively fractionated $\delta^{53}$Cr value of −0.246‰ reported by [19], and the most negatively fractionated value of −0.21 ‰ in the data set published [1,2], while the isotopically most positively fractionated value in our data set (+0.159‰) is somewhat higher than the hitherto highest reported $\delta^{53}$Cr value (+0.068‰) for global chromitites from ophiolite complexes and layered intrusions [1,2].

The diagrams of $\delta^{53}$Cr values versus Cr/(Cr + Al) ratios depict better defined negative trends for chromitites from large deposits such as those from Othrys–Vourinos and Skyros (Figure 6a), than for chromitites from smaller occurrences. Given that the former chromitites are characterized by small degrees of fractionation of the respective parental melts, as exemplified by the low Pd/Ir ratios (Table 1), the observed correlation trends may point to the lowering of $\delta^{53}$Cr values in the melts alongside with the partial melting degree of the respective magma sources. In addition, the most evolved samples from the lower cumulate sequence of the Vourinos ophiolite complex (sample 10) has higher Cr isotope compositions, along with higher Fe contents, compared to the mantle chromitites (represented by Voidolakkos, Kondro and Kissavos (Table 1), that may suggest a coupling of increasing trends of $\delta^{53}$Cr values with the degree of fractionation of the parent magma. Also, with respect to trace elements, including Ni, Co, Zn, V, Mn, Ti and Ga, defined correlations are only depicted between Ga and V concentrations, and Cr/(Cr + Al) ratios. Negative trends are revealed by the Othrys–Vourinos and Skyros chromitites, both of which including high-Cr and high-Al ores (Figure 4). Considering the low Pd/Ir ratios shown by those chromitites (Table 1), these negative trends could be attributed to compositional variations in the parent magmas, rather than to changes inherent during fractionation of these melts.

### 5.1.2. Post-Magmatic Processes

In general, the lack of any clear correlation between major and trace elements, and $\delta^{53}$Cr values may be due to post-magmatic processes (dehydration, serpentinization, metamorphism), which may modify the primary magmatic compositional trends recorded by chromite and silicates (depending on the chromite/silicate ratio), during an extended period of deformation (ductile asthenospheric mantle flow to shallow crustal brittle deformation) [68,72–80]. With respect to the Rhodope massif, on the basis of major-, minor- and trace-element compositions of chromite and Re–Os isotopic compositions of magmatic PGM on the Dobromirtsi chromitites, a long-term history involving reworking of the ancient Archean mantle, during the Phanerozoic, has been suggested [78–82]. Geochemical and mineral chemistry data demonstrated a complex interplay of substitutions, related to the ability of fluids to infiltrate the chromite and the extent of the interaction between pre-existing cores and rims, and that during metamorphism minor and trace elements in chromite can be strongly modified [68]. Also, a sequence of petrological, deformation and redox events has been described for the upper-mantle rocks of the Othrys ophiolite complex. These events are seen to have affected the chromitite bodies during their obduction, emplacement and subsequent alteration [36]. The compositional variations in

the Skyros chromitites in particular [83], may have been caused by redox changes during multistage transformations in the Othrys chromites. Cr oxidation and reduction processes accompanying these deformational events may have caused the variations of Cr isotope signatures measured in those chromitites. (Table 1; Figures 4 and 6).

The variations of $\delta^{53}$Cr signatures in the high-temperature magmatic environment in which the chromitites from the Balkan peninsula were formed, seems to be in a good agreement with the variations measured in some metamorphic Cr-rich minerals such as uvarovites, Cr-tremolite, Cr-diopside, Cr-pyrope. Some of the positively fractionated $\delta^{53}$Cr signatures of these phases have been interpreted as a consequence of heavy Cr isotope enrichments caused by metamorphism [2,21]. Moreover, the investigation of Cr isotopic composition of different types of mantle xenoliths from diverse geological settings (fertile to refractory off-craton spinel and garnet peridotites, pyroxenite veins, metasomatised spinel lherzolites and associated basalts) revealed $\delta^{53}$Cr values of peridotites which also span a range from negatively to positively fractioned values of −0.51 to +0.75‰, and a slightly negative correlation between $\delta^{53}$Cr and $Al_2O_3$ and CaO contents for most mantle peridotites [23].

### 5.2. The Role of Sulphides on the $\delta^{53}$Cr Signatures

The most positively fractionated $\delta^{53}$Cr values are depicted by all massive chromitite samples from the large Othrys complex, and by a few samples from small chromite occurrences (Table 1). Chromitites in certain ophiolite complexes, such as Othrys, Lemesos, Cyprus, Shetland, Bulqiza, Oregon, Moa-Baracoa and elsewhere are associated with Fe–Ni–Cu-sulphide mineralization with dominant minerals pyrrhotite, chalcopyrite and pentlandite, hosted in mantle serpentinized peridotites and ultramafic rocks of the cumulate sequence of the ophiolite complexes [9,17,52,53,62,63]. Although an originally primary magmatic origin for these sulphides is not precluded, characteristics of the highly transformed ore at the Eretria chromite deposit (Greece) may indicate that the original magmatic features have been overprinted and that metals were released from the host rocks by a low-level hydrothermal circulation process [52]. Therefore, in addition to a potential release of Cr from the chromite ore affected by post-magmatic processes, such as brittle deformation and fluid circulation, the less alkaline environment which would be controlled by the dissolution of sulphides could be potentially important for the control of redox-mediated Cr mobilizations and accompanying changes in $\delta^{53}$Cr signatures.

### 5.3. The Potential Role of Abiotic Methane ($CH_4$)

A salient feature of the presented data that potentially could explain the rather large compositional and isotope variations, could be the presence of graphite (Figure 3), and the observation of methane [54] occurring in fluid inclusions of fracture-filling minerals, along with the presence of chlorite and serpentine. These features are especially present in the Othrys chromitites, which also show the most positively fractionated $\delta^{53}$Cr values (Table 1; Figure 5). Recently, the investigation of microdiamonds and graphite in chromite from ophiolite-type chromitites hosted in the Tehuitzingo serpentinite (southern Mexico), have been attributed a secondary origin and process related to the retrograde evolution of the respective chromitite. These processes apparently took place at relatively low temperatures (520–670 °C), at low pressures and in shallow depths [69]. The presence of minerals indicative of super-reducing phases, such as graphite-like, Fe–Ni–Cr alloys, awaruite ($Ni_3Fe$), and heazlewoodite ($Ni_3S_2$) [44,45,82,83], in the chromitite ores from the islands of Othrys and Skyros may be suggestive of the introduction of continental crust derived, reduced C–O–H fluids, during post-magmatic alteration processes. These were probably facilitated during brittle deformation stages in the shallow crust and in the subducted oceanic slab [69,84,85]. The association of graphite with sperrylite and sulphides in magnetite ore from the Skyros ophiolite complex has also been attributed to its deposition from a $CO_2^-$ and $CH_4^-$-dominated reducing fluid at low oxygen fugacity at relatively low temperatures (500–300 °C) during serpentinization [58]. In addition, methane is widespread in microfractures and porous serpentine- or chlorite-filled veins [54]. The widespread occurrence of

methane and Ni–V–Co phosphides in the Othrys chromitites [54,84,85] and graphite in the Skyros island ophiolite (Figure 3) seems to be compatible with the positively fractionated $\delta^{53}$Cr values (Table 1), features pointing to the circulation of reducing fluids, probably during serpentinization, which facilitated the back-reduction of mobilized Cr(VI) fractions to secondary immobile Cr(III)-bearing mineral phases [32,86].

Therefore, the relatively elevated, slightly heavy Cr isotope signatures recorded in chromitites may be the result of a combination of (a) fractionation during magmatic processes, i.e., degree of partial melting and differentiation of the parent magma, and (b) multi-stage post-magmatic overprinting, which seem to depend on the size of the respective chromitite occurrence (in terms of tonnage of chromite) and on a probable relationship between the size and degree of shielding towards alteration of the chromitite bodies.

*5.4. Weathering of Cr-Bearing Minerals and Environmental Significance*

Contamination of groundwater and soil by heavy metals is becoming a serious threat to our environments worldwide. The presence of harmful Cr(VI) in soils and in turn in groundwater may be related to natural processes or/and human activities, such as transfer of weathered material from rocks and primary raw materials, wastes and/or application of large amounts of fertilizers and pesticides for a long time in cultivated areas. The European Soil Data Centre has established a link between the effects of metal bio-availability and metal bio-accumulation, and human health and negative impacts on ecosystems [25–30,87–96]. Dose-dependent differences in toxicities of elements, the particle size, and the oxidation state require serious consideration in health-risk assessments [90]. A very common source of salts in irrigated soils is the irrigation water itself, affected by sea water in low-lying areas along the coast. The salinization of shallow aquifers may have a major effect on plant/crop growth and toxicity. The use of multidisciplinary methods in the study of ecosystem processes in response to groundwater and soil system has great potentials for sustainable developments [87–96].

The application of Cr isotope measurements to evaluate Cr(VI) contamination in groundwater and rock leachates from the central Evia and Assopos basin in Greece has revealed positively fractionated $\delta^{53}$Cr values ranging from +0.56 to +0.96‰ in water leachates of ultramafic rocks, which are comparable to those from Central Europe [26–28]. These results imply oxidative mobilization of Cr(VI) from the ultramafic host rocks, and successive back-reduction of so mobilized Cr(VI) fractions [29]. The widespread distribution of ophiolite complexes in Europe and in orogenic zones elsewhere around the globe, emphasizes the need for further more detailed studies addressing the variation of $\delta^{53}$Cr values in Cr-bearing rocks and ores.

In summary, the original geochemical compositions of chromitites may have been modified by subsequent, post-magmatic processes taking place at shallow mantle depths [35–38,56,57,68,69,72–89,97]. The wide range of $\delta^{53}$Cr values, from positive, slightly negative to the most negatively fractionated signatures, even present in individual relatively large chromitite deposits, together with negative trends observed between $\delta^{53}$Cr values and Cr/(Cr + Al) ratios in chromitites, may reflect a control of Cr isotope compositions in chromitites by degrees of partial melting and by magma fractionation. This is best exemplified by the high-Al chromitites from the cumulate sequence of the Vourinos complex. In addition, post-magmatic redox cycling of Cr-isotopes may occur in response to brittle deformation of chromitites and the subsequent interaction of the Cr-bearing rocks with oxidative fluids, also potentially leading to the formation of Fe-chromite through substitution of Cr, Al and Mg by Fe(III) and Fe(II). The formation of Cr-bearing minerals, such as serpentine, chlorite, and some garnets, and the concomitant release of Cr(VI)-bearing fluids, under alkaline conditions, may take place during serpentinization of respective ore bodies. Mobilized Cr(VI) bearing fluids then are prone to back-reduction promoted by the reduced environment in serpentinized ultramafic rocks, causing the formation of graphite, secondary Fe–Cr-bearing silicates, and other secondary alteration phases that potentially can carry a positively fractionated Cr isotope signal.

## 6. Conclusions

The $\delta^{53}$Cr values presented herein for chromitites from the Balkan Peninsula offer an opportunity to study the effects of Cr isotope fractionations that potentially could be related to primary magmatic processes and to compare these with compositional trends eventually depicted by the respective chromite deposits and chromite occurrences.

A modification of the magmatic control on the geochemical characteristics of chromitites is revealed by positively fractionated $\delta^{53}$Cr values in ophiolite zones affected by brittle deformation/metamorphism and characterized by the presence of secondary epigenetic Cr-bearing minerals, formed by the circulation of reducing fluids carrying abiotic methane.

Chromitites from the Balkan Peninsula depict a wide range in $\delta^{53}$Cr values. Signatures range from positively-, to slightly negatively fractionated $\delta^{53}$Cr values, even in individual relatively large deposits. Positively fractionated $\delta^{53}$Cr values of all chromitite samples from Othrys and of high-Al chromitites from Skyros, and a negatively correlated trend between $\delta^{53}$Cr and Cr/(Cr + Al), may reflect a control of $\delta^{53}$Cr by degree of partial melting and by magma fractionation. This is best exemplified by high-Al chromitites from the cumulate sequence of the Vourinos complex.

The application of Cr isotopes to the evaluation of Cr(VI) contamination in groundwater and rock leachates relies on oxidative mobilization of Cr(VI) and successive back-reduction to C(III). The widespread distribution of significant natural Cr-sources in the Balkan peninsula, enhanced by industrial activities involving Cr in their processes, present a latently dangerous framework which is prone to have negative effects on the environment, particularly with respect to hexavalent Cr toxicity. Detailed studies addressing the variation of $\delta^{53}$Cr values in Cr-bearing rocks, ores, soils, surface run-off and groundwater are required to understand the complicated mutual interplay between original natural Cr sources and the weathering-induced release of Cr into the environment.

**Author Contributions:** Conceptualization, R.F. and M.E.-E.; Methodology, M.E.-E., R.F. and I.M.; R.F. provided the chromium isotope analyses and their validation. M.E.-E. and I.M. provided the field information and performed the other analyses. All authors (M.E.-E., R.F. and I.M.) contributed to the elaboration, interpretation of the data and to the writing of the manuscript. All authors have read and agreed to the published version of the manuscript.

**Funding:** This research was funded by the National and Kapodistrian University of Athens (NKUA) (Grant No. KE_11730).

**Acknowledgments:** Many thanks are expressed to the two anonymous reviewers for their constructive criticism and suggestions on an earlier draft of the manuscript. Vassilios Skounakis in thanked for his assistance with SEM/EDS analysis of the samples. RF thanks Toby Leeper for always keeping the mass spectrometers at IGN in perfect running condition, and Toni Larsen for help in the ion chromatographic separation of the samples. This paper is dedicated to the memory of Maria Zhelyaskova-Panayiotova, University of Sofia, who provided the chromitite samples from Bulgaria, and passed away in May 2018.

**Conflicts of Interest:** The authors declare no conflict of interest.

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
