# Peer review of "Factors Controlling the Chromium Isotope Compositions in Podiform Chromitites"

_minerals, doi:10.3390/min10010010_

Round 1

Reviewer 1 Report

see file

Author Response

Comment: Minerals Manuscript Number: 653997 Title: Factors controlling the chromium isotope compositions in podiform chromitites Article Type: Research Paper Keywords: Chromium isotopes; chromitites; ophiolites; Balkan Pensula The manuscript Minerals-653997 provides a geochemical study on Cr stable isotope data (δ53Cr values) with major and trace element composition, from the Balkan Peninsula, aiming to provide an overview of the compositional variations of δ53Cr values in ophiolite-hosted chromitites and delineate geochemical constraints controlling the composition of chromitites.

I recommend it for publication after a major revision. The main trouble points are: Comments Introduction Is the main problem of this work, it is short and all the main references to Chromium and Chromium isotopes are missing.

some examples are: Slejko, F. F., Petrini, R., Lutman, A., Forte, C., & Ghezzi, L. (2019). Chromium isotopes tracking the resurgence of hexavalent chromium contamination in a past-contaminated area in the Friuli Venezia Giulia Region, northern Italy. Isotopes in environmental and health studies, 55(1), 56-69.

Qin, L., & Wang, X. (2017). Chromium isotope geochemistry. Reviews in Mineralogy and Geochemistry, 82(1), 379-414.

Reply: In the section of Introduction a paragraph was added concerning main references to Cr and Cr- isotopes and the suggested references are included now (in red color)

Comment: A brief outline of characteristics of ophiolites and hosted chromitites.

Reply: Characteristics of ophiolites and hosted chromitites are given now.

Comment: The geological map is missing.

Reply: A simplified geological map is given now.

Comment: Discussion 5.4. Weathering of Cr-bearing minerals and environmental significance Also in this case, much of the more recent bibliography is not mentioned. See for example: Apollaro, C., Marini, L., Critelli, T., Barca, D., Bloise, A., De Rosa, R., Liberi, F., Miriello, D., 2011. Investigation of rock-to-water release and fate of major, minor, and trace elements in the metabasaltserpentinite shallow aquifer of Mt. Reventino (CZ, Italy) by reaction path modeling. Appl. Geochem. 26, 1722–1740. https://doi.org/10. 1016/j.apgeochem.2011.04.028.

Apollaro, C., Fuoco, I., Brozzo, G., De Rosa, R., 2019. Release and fate of Cr (VI) in the ophiolitic aquifers of Italy: the role of Fe (III) as a potential oxidant of Cr (III) sup- ported by reaction path modelling. Sci. Total Environ. 660, 1459–1471. https://doi. org/10.1016/j.scitotenv.2019.01.103.00

Reply: In the section of the Discussion two paragraphs were added on the soil and groundwater contamination by Cr(VI), including all suggested references.

Comment: The English poor

Reply: English was improved.

Comment: Figures: In the Figure 4 and in Figure 6 , R should be replaced with R2

Reply: R was corrected as r in Figures 4 and 6. It is clarified that r is to express the degree of relationship between two variables. As far as we are aware for multiple linear regression R is computed, but then it is difficult to explain, because we have multiple variables. That is why R square is a better term. In addition, we do not use R square because it would be inconsistent with the negative correlations of the Figs 4 and Fig 6a, since square values are positive.

Comment: Reformulate the discussions and conclusions taking into account the recommended works in the bibliography References Add these references: Slejko, F. F., Petrini, R., Lutman, A., Forte, C., & Ghezzi, L. (2019). Chromium isotopes tracking the resurgence of hexavalent chromium contamination in a past-contaminated area in the Friuli Venezia Giulia Region, northern Italy. Isotopes in environmental and health studies, 55(1), 56-69. Qin, L., & Wang, X. (2017). Chromium isotope geochemistry. Reviews in Mineralogy and Geochemistry, 82(1), 379-414. Apollaro, C., Marini, L., Critelli, T., Barca, D., Bloise, A., De Rosa, R., Liberi, F., Miriello, D., 2011. Investigation of rock-to-water release and fate of major, minor, and trace elements in the metabasaltserpentinite shallow aquifer of Mt. Reventino (CZ, Italy) by reaction path modeling. Appl. Geochem. 26, 1722–1740. https://doi.org/10. 1016/j.apgeochem.2011.04.028. Apollaro, C., Fuoco, I., Brozzo, G., De Rosa, R., 2019. Release and fate of Cr (VI) in the ophiolitic aquifers of Italy: the role of Fe (III) as a potential oxidant of Cr (III) sup- ported by reaction path modelling. Sci. Total Environ. 660, 1459–1471. https://doi. org/10.1016/j.scitotenv.2019.01.103

Reply: In the sections of the Discussion and Conclusions the soil and groundwater contamination by Cr(VI) are discussed and the suggested references are included now.

Many thanks for the consideration of our manuscript and the constructive comments.

Reviewer 2 Report

Dear authors,

Your paper presents some very interesting new data about Greek chromitites.

Although your work is known and you are very experienced on this subject, nevertheless you have failed to provide sufficient background and include all relevant references on the mineral chemistry of these chromitites and their accessory minerals (e.g. heazlewoodite, awaruite etc.) by totally excluding from your references a lot of journal and conference papers recently published.

Please take care of this significant issue in order to meet the criteria for publication of this very good work.

Regards,

Author Response

Comment:

Dear authors,

Your paper presents some very interesting new data about Greek chromitites.

Although your work is known and you are very experienced on this subject, nevertheless you have failed to provide sufficient background and include all relevant references on the mineral chemistry of these chromitites and their accessory minerals (e.g. heazlewoodite, awaruite etc.) by totally excluding from your references a lot of journal and conference papers recently published.

Please take care of this significant issue in order to meet the criteria for publication of this very good work.

 Regards,

Reply: In the sections of (a)Introduction,  (b) Characteristics of ophiolites, (c) Weathering of Cr-bearing minerals and environmental significance, (d) and Discussion main references to Cr and Cr- isotopes are included (in red color) and are discussed now.

Many thanks for the consideration of our manuscript and the constructive comments.

Round 2

Reviewer 1 Report

it's ok

Author Response

Dear reviewer,

Thank you very much for your constructive comments 

All the best

Maria Economou-eliopoulos

Reviewer 2 Report

Dear authors,

although your manuscript seems to be improved, nevertheless you have failed to include in your manuscript all published articles in this subject. In order to facilitate your effort, I am highlighting the research groups and their supervisors that have recently published relevant articles:

1) University of Thessaloniki, Professor Filippidis

2) University of Patras, Professor Chatzipanagiotou

3) University of Milano, Professor Grieco

Please make a thorough search in the published literature of the above mentioned groups and include the relevant publications and their findings (compared to your results) in your manuscript.

Regards

Author Response

Comment:

Dear authors,

although your manuscript seems to be improved, nevertheless you have failed to include in your manuscript all published articles in this subject. In order to facilitate your effort, I am highlighting the research groups and their supervisors that have recently published relevant articles:

1) University of Thessaloniki, Professor Filippidis

2) University of Patras, Professor Chatzipanagiotou

3) University of Milano, Professor Grieco

Please make a thorough search in the published literature of the above mentioned groups and include the relevant publications and their findings (compared to your results) in your manuscript.

Regards

Reply: After a thorough search in the published literature of the suggested authors we selected and are included all references which are related to the topic and the aim of this manuscript.

Many thanks for the consideration our manuscript again

Regards

Maria Economou-Eliopoulos

Round 3

Reviewer 2 Report

Dear authors,

In order to properly refer to relevant works in the frame of academic ethics, please also take into account the following publications which you should already have found and cite:

-Iron oxidation state in Greek Cr-ore minerals

Tzamos et al.

-Mg-Fe diffusivity patterns in sub-continental and ophiolite mantle chromitites

Bussolessi et al.

-Processes of primary and re-equilibration mineralization affecting chromitite ore geochemistry within the Vourinos ultramafic sequence, Vourinos Ophiolite (West Macedonia, Greece)

Grieco et al.

-Metallogeny of the Chrome Ores of the Xerolivado-Skoumtsa Mine, Vourinos Ophiolite, Greece: Implications on the genesis of IPGE-bearing high-Cr chromitites within a heterogeneously depleted mantle section

Tzamos et al.

-MINERAL CHEMISTRY AND FORMATION OF AWARUITE AND HEAZLEWOODITE IN THE XEROLIVADO CHROME MINE, VOURINOS, GREECE

Tzamos et al.

-Major and minor element geochemistry of chromite from the Xerolivado-Skoumtsa mine, Southern Vourinos: Implications for chrome ore exploration

Tzamos et al.

Should you include these references, there is no other comment from my side and I will propose the publication of your very good paper.

Regards,

Author Response

Reply:

Thank you very much for the consideration of our manuscript. However, I am very sorry to say that in our opinion all relevant publications in the frame of academic ethics have been cited. The  suggested references have all as topic the Xerolivado-Skoumtsa mine. Although it belongs to the Vourinos complex, it is not included in the Table 1: Results of the studied chromitites herein.

Nevertheless following the your suggestion the following references were added:

Filippidis, A. (1997) Chemical variation of chromite in the central sector of xerolivado chrome mine of Vourinos, Western Macedonia, Greece. NJ Miner, 1997, 8, 354–370

Tzamos, E.; Filippidis, A.; Rassios, A.; Grieco, G.; Michailidis, K.; Koroneos, A.; Gamaletsos, P.N. Major and minor element geochemistry of chromite from the Xerolivado-Skoumtsa mine, Southern Vourinos: implications for chrome ore exploration. J. Geochem. Explor., 2016, 165, 81–93.

Tzamos, E.; Kapsiotis, A.; Filippidis, A., Koroneos, A.; Grieco, G., Rassios, A.E.; Godelitsas, A. Metallogeny of the Chrome Ores of the Xerolivado-Skoumtsa Mine, Vourinos Ophiolite, Greece: implications on the genesis of IPGE-bearing high-Cr chromitites within a heterogeneously depleted mantle section. Ore Geol. Rev. 2017, 90, 226–242.

Grieco, G.; Bussolesi, M.; Tzamos, E.; Rassios, A.E.; Kapsiotis, A. Processes of primary and re-equilibration mineralization affecting chromitite ore geochemistry within the Vourinos ultramafic sequence, Vourinos ophiolite (West Macedonia, Greece). Ore Geol. Rev. 2018, 95, 537–551.

In our opinion the following references are unrelated to our manuscript and are not necessary:

Tzamos E., Filippidis A., Michailidis K., Koroneos A., Rassios A., Grieco G., Pedrotti M. and Stamoulis K. (2016) MINERAL CHEMISTRY AND FORMATION OF AWARUITE AND HEAZLEWOODITE IN THE XEROLIVADO CHROME MINE, VOURINOS, GREECE

-Iron oxidation state in Greek Cr-ore minerals

Tzamos et al.

-Mg-Fe diffusivity patterns in sub-continental and ophiolite mantle chromitites

Bussolessi et al.

Regards
